# On the Effect of Imperfect Reference Signal Phase Recovery on Performance of PSK System Influenced by TWDP Fading

**DOI:** 10.3390/e25091341

**Published:** 2023-09-15

**Authors:** Goran T. Djordjevic, Dejan N. Milic, Bata Vasic, Jarosław Makal, Bane Vasic

**Affiliations:** 1Faculty of Electronic Engineering, University of Nis, 18104 Nis, Serbia; dejan.milic@elfak.ni.ac.rs (D.N.M.); bata.vasic@elfak.ni.ac.rs (B.V.); 2Faculty of Electrical Engineering, Białystok University of Technology, 15-351 Białystok, Poland; j.makal@pb.edu.pl; 3Department of Electrical & Computer Engineering, University of Arizona, Tucson, AZ 85721-0104, USA; vasic@arizona.edu

**Keywords:** fading channel, error probability, wireless communications, simulations

## Abstract

We examine the effects of imperfect phase estimation of a reference signal on the bit error rate and mutual information over a communication channel influenced by fading and thermal noise. The Two-Wave Diffuse-Power (TWDP) model is utilized for statistical characterization of propagation environment where there are two dominant line-of-sight components together with diffuse ones. We derive novel analytical expression of the Fourier series for probability density function arising from the composite received signal phase. Further, the expression for the bit error rate is presented and numerically evaluated. We develop efficient analytical, numerical and simulation methods for estimating the value of the error floor and identifying the range of acceptable signal-to-noise ratio (SNR) values in cases when the floor is present during the detection of multilevel phase-shift keying (PSK) signals. In addition, we use Monte Carlo simulations in order to evaluate the mutual information for modulation orders two, four and eight, and identify its dependence on receiver hardware imperfections under the given channel conditions. Our results expose direct correspondence between bit error rate and mutual information value on one side, and the parameters of TWDP channel, SNR and phase noise standard deviation on the other side. The results illustrate that the error floor values are strongly influenced by the phase noise when signals propagate over a TWDP channel. In addition, the phase noise considerably affects the mutual information.

## 1. Introduction

Beyond 5G networks should satisfy many Quality-of-Service (QoS) requirements related to high data rates, reliability, energy efficiency and security, as well as low latency and wide radio coverage [1]. In order to achieve high spectral efficiency, high-order modulations are applied. However, as the order of modulation increases, transmission of information becomes more sensitive to both channel impairments and receiver hardware imperfections [2]. In general, signal detection can be coherent or incoherent. The application of coherent detection is motivated by potential energy gains. However, in order to realize coherent detection, it is necessary to regenerate the phase of the reference carrier. The reference carrier phase can be extracted using a phase-locked loop (PLL) [2,3,4,5,6]. The regenerated phase should follow the random signal phase fluctuations generated due to multipath propagation. Even in a channel with additive white Gaussian noise (AWGN), the extraction of the reference carrier phase is not ideal [6]. In other words, there is a difference between the signal phase at the receiver input and the regenerated carrier phase. The difference between these two phases is called the phase error or phase noise. This phase error is a random process and is known to lead to the significant degradation of the system performance [2,3,4,5,6].

### 1.1. Literature Review

In wireless channels where, in addition to thermal noise, there is also multipath signal propagation, the problem of imperfect phase estimation from the received signal becomes even more actual. A number of distinct statistical models have been proposed so far in order to represent the effect of multipath fading, such as Rayleigh, Rice, Nakagami-*m*, Nakagami-*q*, etc. [2]. When multipath fading follows Nakagami-*m* distribution, its influence on error probability in conjunction with phase noise is considered in [3], while shadowed multipath fading is analyzed in a similar scenario in [4].The influence of the phase noise on the error probability when multipath fading is modeled by Nakagami-*m* distribution was considered in [3], while the effect of imperfect reference signal extraction on the error probability over a shadowed multipath fading was analyzed in [4]. Smadi and Prabhu in [5] presented a novel treatment in estimating the effect of phase noise on error performance of binary phase-shift keying (BPSK) and quaternary phase-shift keying (QPSK) with equal gain combined under Rayleigh, Rician and Nakagami-*m* fading environment. These papers show that communication systems performance can be significantly impaired by imperfect reference carrier recovery in adition to multipath fading effects alone, when signals propagate over fading channels of Rayleigh, Rician, Nakagami-*m* and shadowed Nakagami-*m* type.

Recently, several papers suggested models for mm-Wave channels [1,7,8,9,10,11,12,13]. All of these distributions are from the group of fluctuating multiple-ray models [1]. In a case when two dominant line-of-sight (specular) waves reach the receiver, alongside the diffuse wave components of the transmission, there is a suitable fading model of so-called Two-Way Diffuse-Power (TWDP) that was initially proposed by Durgin et al. in [7]. In the paper that introduces the TWDP model, the authors also present an approximate expression for probability density function (PDF) regarding the statistical variations of the signal envelope. The fading model has been continuously studied by other researchers [8,9,10,11,12,13]. In [8], Kim et al. discuss shortcomings of approximate expression from [7], and derive exact and suitable approximate expressions for determining bit error rate (BER) in cases when the BPSK signal is detected after being transmitted over a TWDP channel while retaining large SNR values. In [9,10], Rao et al. develop novel expressions for system performance metrics and present an interesting result showing that the received signal envelope conforming to the TWDP fading model encompasses a closed-form moment generation function (MGF). The authors of [11,12] propose a novel aspect on TWDP channel model parameters that differs in comparison with parametrization formerly used in [7,8,9,10]. This parametrization is actually based on formula originally derived in [14]. An experimental analysis has shown recently that this fading model is well suited for signal description in outdoor and indoor propagation environments in the 60 GHz range [13].

### 1.2. Contribution

To the best of our knowledge, all previous works considering the TWDP fading model are based on the assumption of ideal reference carrier phase extraction [7,8,9,10,11,12,13]. The results obtained in such a way should be considered optimistic, in general. In this paper, we consider the effect of imperfect phase recovery of reference signal on system performance when signal propagates over a TWDP channel. The reference signal recovery can be performed from modulated received signal or from pilot signal. Both ways for reference signal phase recovery over TWDP channel are open problems. For illustrative purposes, we are focusing on reference signal recovery from the pilot signal, where the loop bandwidth is much lesser than the channel bandwidth [3,4,5]. The signal phase at the voltage-controlled oscillator (VCO) output does not perfectly follow the phase of the signal being detected. Generally, a stochastic difference exists between the incoming signal phase and the signal phase at the VCO output. The phase noise is modeled by Tikhonov distribution according to [2,3,4,5,6]. Our goal in this paper is to determine the impact of phase noise on the BER performance and mutual information [15,16] when multilevel phase-shift keying (PSK) signal is transmitted over a TWDP channel. We estimate the error probability in three different ways. Firstly, we present the Fourier Series Method (FSM) for estimating the symbol-error rate (SER) in detecting multilevel PSK signals over a TWDP channel. This method is based on representing the PDF of the composite received signal phase in terms of Fourier series with coefficients depending on the channel conditions. This method is presented in [2] for the case when multipath fading is described by Nakagami-*m* distribution. The main challenge in application of FSM is to express the Fourier series coefficients in appropriate analytical form for given propagation environment. Here, we derive new analytical formulas for Fourier coefficients, which provide us with the possibility to determine the SER for any value of the number of phase levels (denoted by *M*) and for all numerical values of TWDP channel parameters encountered in practice. Secondly, we provide formulas for numerically calculating the BER when detecting BPSK and QPSK signals followed by explanations of numerical evaluations. Thirdly, numerical values are verified by independent Monte Carlo simulations. Further, by using Monte Carlo simulations, we calculate mutual information for modulation orders of two, four, and eight. We connect error rate and mutual information with the standard deviation of phase noise, channel parameters and SNR in the case when reference signal phase recovery is performed from the pilot signal. The results presented here for QPSK correspond directly to the fourth-order quadrature amplitude modulation (4QAM) format suggested for 5G and beyond networks [17]. We show that our results can be reduced to already published results from the literature in the special case when the assumption is introduced that the reference signal phase recovery is perfect [9].

### 1.3. Structure

The paper is organized in the following way. In Section 2, we describe the model of the system containing a transmitter, a channel, and a receiver. In Section 3, we explain the evaluation of error probability and mutual information, while Section 4 presents numerical and simulation results. Concluding remarks are emphasized in Section 5.

## 2. System Model

In this Section, we describe the system model (Figure 1) containing a transmitter, a channel, and a receiver. The channel model and receiver imperfections are described in greater detail.

### 2.1. Transmitter

The information bits are mapped into symbols using the Gray code. If we denote the order of digital phase modulation by *M*, then one symbol contains $\log_{2}M$ bits. The symbols are then written to the interleaver and read out from it. The interleaver is used in order to make fading values over the channel to be mutually uncorrelated over the adjacent symbols. Also, it is assumed that the values the thermal noise during the duration of one symbol are uncorrelated with the values of this noise during the duration of the next symbol, which also applies to the influence of the phase noise. In other words, by using interleaving at the transmitter and complementary de-interleaving in the receiver, the signal is transmitted over a memoryless channel. The interleaver output symbols are fed to the MPSK modulator. Over the duration of one information symbol, the modulator outputs a signal of the following form: $s_{i}=A\;e^{\;j\phi_{n}}$, where the signal amplitude *A* is constant and the modulated phase has a value from the set $\phi_{n}\in \left\{0,2\pi /M,\cdots 2\pi \left(M-1\right)/M\right\}$, where *M* is the number of phase levels [2].

### 2.2. Channel

We assume that a wireless channel can be established between a transmitter and a receiver, and the signal is transmitted over this channel. During the signal transmission, there are multiple copies of the transmitted EM wave, and they simulataneously excite the receiver antenna. The TWDP model is general and it assumes two dominant line-of-sight (LOS) components and diffuse components. The two specular components have amplitudes that are constant, and their phases are distributed uniformly over the interval from 0 to 2π. We denote the amplitudes of these specular components by $V_{1}$ and $V_{2}$, while their phases are denoted by $\Psi_{1}$ and $\Psi_{2}$. The scattering component statistics correspond to a Rayleigh distribution, i.e., it consists of two Gaussian-distributed in-phase and quadrature components having zero mean and same standard deviations of $\sigma_{F}$. These random in-phase and quadrature components are denoted by $x_{F}$ and $y_{F}$, respectively. The resulting complex signal envelope at the receiver is [7]
(1)$$V_{r}=V_{1}\;e^{\;j\Psi_{1}}+V_{2}\;e^{\;j\Psi_{2}}+x_{F}+jy_{F}.$$
The complex fading can be presented in terms of the envelope *r* and argument θ as
(2)$$V_{r}=r\;e^{\;j\theta }.$$
The resulting envelope is given by
(3)$$r=\sqrt{{\left(V_{1}\cos\Psi_{1}+V_{2}\cos\Psi_{2}+x_{F}\right)}^{2}+{\left(V_{1}\sin\Psi_{1}+V_{2}\sin\Psi_{2}+y_{F}\right)}^{2}}.$$

The PDF of the fading envelope can be expressed in different forms. An approximation is given by ([7], Equation (Equation 4)). However, some shortcomings of this approximation were emphasized in [8]. By using an analogous with the situation when there are a useful signal, co-channel interference and Gaussian noise, an infinite series form of PDF was presented in [11], where the authors introduced the novel way for parametrization. Namely, this fading model can be described using two parameters denoted by *K* and Γ. The value of parameter *K* represents the power of the specular components-to-power of the diffuse component, while parameter Γ is the ratio between the amplitudes of specular components. These two parameters are defined as [11]
(4)$$K=\left(V_{1}^{2}+V_{2}^{2}\right)/\left(2\sigma_{F}^{2}\right),\;\Gamma =V_{2}/V_{1}.$$

By varying the value of the basic parameter denoted by *K* in our work, it is possible to make specular components dominant compared with diffuse components, and vice versa. With increasing the value of parameter *K*, the specular components become dominant in comparison to the diffuse component. For example, in the terrestrial mobile links, parameter *K* has values that typicaly range from 0 dB to 15 dB [7,8,9,10,11,12]. In addition, by varying the value of parameter Γ, it is possible to scale the values of amplitudes of two specular components. The value of parameter Γ belongs to the interval between 0 and 1. When Γ=0, it follows that the one specular component is equal to zero, while when Γ=1, it follows that the amplitudes of specular components are equal. Instead of parameter Γ, the authors of [7,8,9,10] used parameter Δ defined as $\Delta =2V_{1}V_{2}/\left(V_{1}^{2}+V_{2}^{2}\right)$.

Without losing generality, we introduce the assumption that the mean of squared envelope value equals one, i.e.,
(5)$$\bar{r^{2}}=V_{1}^{2}+V_{2}^{2}+2\sigma_{F}^{2}=1.$$
From (4) and (5), it follows that
(6)$$V_{1}=\sqrt{K/\left(\left(1+\Gamma^{2}\right)\left(1+K\right)\right)},\;V_{2}=V_{1}\;\cdot \;\Gamma ,$$
(7)$$\sigma_{F}=\sqrt{1/\left(2(1+K)\right)}.$$

Based on the derivation approach from [11], the PDF of the fading envelope can be expressed in the form of
(8)$$p_{R}\left(r\right)=\frac{r}{\sigma_{F}^{2}}\;e^{-\frac{r^{2}+V_{1}^{2}+V_{2}^{2}}{2\sigma_{F}^{2}}}\left\{I_{0}\left(\frac{rV_{1}}{\sigma_{F}^{2}}\right)I_{0}\left(\frac{rV_{2}}{\sigma_{F}^{2}}\right)I_{0}\left(\frac{V_{1}V_{2}}{\sigma_{F}^{2}}\right)+2\sum_{m=1}^{+\infty }{\left(-1\right)}^{m}\;I_{m}\left(\frac{rV_{1}}{\sigma_{F}^{2}}\right)I_{m}\left(\frac{rV_{2}}{\sigma_{F}^{2}}\right)I_{m}\left(\frac{V_{1}V_{2}}{\sigma_{F}^{2}}\right)\right\},$$
where $I_{\nu }\left(\;\cdot \;\right),\nu =0,1,\cdots$ denotes the modified Bessel function of the first kind and order ν ([18], Equation (8.431)).

### 2.3. Receiver

The signal at the receiver input can be presented as
(9)$$s_{0}=r\;e^{\;j(\phi_{n}+\theta )}+n,$$
where *r* is the fading envelope, θ is a random signal phase due to multipath propagation, and *n* represents narrowband Gaussian noise. The in-phase and quadrature components of noise *n* are random signals with Gaussian PDF whose mean values are zero, and standard deviations are identical and denoted by σ.

The received signal random phase variations due to multipath fading should be detected and removed during the demodulation process. Actually, PLL has the role to follow these phase variations. In practice, the phase extractor in the receiver is not ideal, and there is a difference between received signal phase and estimated phase. The difference between these two phases is called the phase error, which is a random variable. It is well known that in the case when there is only AWGN in the loop circuit, this phase error is a random variable with Tikhonov PDF [3,4,5,6]. In this paper, we assume that the phase error can be modeled by Tikhonov distribution because the reference signal recovery is performed from pilot signal, and the PLL bandwidth is much lesser than the channel bandwidth. It means that the PDF of the phase error can be written in the form of [3,4,5,6]
(10)$$p_{\varphi }\left(\varphi \right)=\frac{e^{\;\rho_{\mathrm{PD}}\cos\varphi }}{2\pi \;I_{0}\left(\rho_{\mathrm{PD}}\right)},\;\left|\varphi \right|\leq \pi ,$$
where $\rho_{\mathrm{PD}}$ is SNR in the PLL circuit. This $\rho_{\mathrm{PD}}$ can be linked to the phase noise standard deviation as $\rho_{\mathrm{PD}}=1/\sigma_{\varphi }^{2}$ [3,4,5,6]. Formula (Equation 10) can be presented in the form of [3,4]
(11)$$p_{\varphi }\left(\varphi \right)=\frac{1}{\pi }+\sum_{n=0}^{+\infty }c_{n}\cos\left(n\varphi \right),\;\left|\varphi \right|\leq \pi ,$$
where coefficients $c_{n}$ are defined as $c_{n}=I_{n}\left(\rho_{\mathrm{PD}}\right)/\left(\pi I_{0}\left(\rho_{\mathrm{PD}}\right)\right)$.

After signal demodulation, the in-phase and quadrature signal components in the upper and lower branches of the receiver can be presented as
(12)$$\begin{array}{cc} s_{x} & =r\cos\left(\phi_{n}+\varphi \right)+n_{x},\\ s_{y} & =r\sin\left(\phi_{n}+\varphi \right)+n_{y}, \end{array}$$
where $n_{x}$ and $n_{y}$ have zero mean value and standard deviation denoted by σ. The detection is performed based on the value of
(13)$$\tan\delta =s_{y}/s_{x}.$$

## 3. Performance Evaluation

Due to the presence of noise and other disturbances like fading and hardware imperfections, the detected bits in the receiver contain errors. In addition, it is well known that noise and other interferences over a channel constraint the mutual information. Because of that, the primary aim is to estimate BER and mutual information in the presence of the previously mentioned effects. In this Section, we offer formulas for analytical and numerical evaluation of BER and present the model for Monte Carlo simulations. After that, we present a procedure for estimating numerical values of the mutual information under given constraints.

### 3.1. Error Probability

#### 3.1.1. Analytical Approach

The conditional PDF of the phase of the composite signal at the receiver input can be presented in the form of
(14)$$p_{\Delta |R}\left(\delta |r\right)=\frac{1}{2\pi }+\sum_{n=0}^{+\infty }a_{n}\left(r\right)\cos\left(n\delta \right),\;\left|\delta \right|\leq \pi ,$$
where coefficients $a_{n}\left(r\right)$ are defined by [3,4]
(15)$$a_{n}\left(r\right)=\frac{1}{n!\pi }\Gamma \left(1+\frac{n}{2}\right){\left(\frac{r}{\sqrt{2}\sigma }\right)}^{\;\;n}\exp\left(-\frac{r^{2}}{2\sigma^{2}}\right){}_{1\;}F_{\;1}\;\left(\frac{n}{2}+1;n+1;\frac{r^{2}}{2\sigma^{2}}\right).$$

Unlike in our previous paper [4] in which the coefficients $a_{n}$ were given as a function of the instantaneous SNR, here, we offer those coefficients as a function of the signal envelope. The average PDF of the composite signal phase can be written as
(16)$$p_{\Delta }\left(\delta \right)=\frac{1}{2\pi }+\sum_{n=0}^{+\infty }b_{n}\cos\left(n\delta \right),\;\left|\delta \right|\leq \pi ,$$
where coefficients $b_{n}$ can be determined based on
(17)$$b_{n}=\int_{0}^{+\infty }a_{n}\left(r\right)p_{R}\left(r\right)\;dr.$$
The result is due to the fact that integrals and countable sums are interchangeable for non-negative functions, such as $a_{n}\left(r\right)p_{R}\left(r\right)$, according to Tonelli’s theorem. Obviously, $a_{n}\left(r\right)$ is nonnegative, and the same is true by definition for any PDF $p_{R}\left(r\right)$ on [0,+∞).

The CDF of the signal envelope can be presented in the form of [19]
(18)$$F_{R}\left(r\right)=\frac{r^{2}\;e^{-\frac{r^{2}}{2\sigma_{F}^{2}}}}{2\sigma_{F}^{2}}\;\sum_{m=0}^{+\infty }\;\frac{{\left(-1\right)}^{m}}{m!}h_{1}^{2m}{}_{2\;}F_{\;1}\;\left(-m,-m;1;h_{2}^{2}\right){}_{1\;}F_{\;1}\;\left(1-m;2;\frac{r^{2}}{2\sigma_{F}^{2}}\right),$$
where $h_{1}^{2}=\frac{V_{1}^{2}}{2\sigma_{F}^{2}}$, $h_{2}^{2}=\frac{V_{2}^{2}}{V_{1}^{2}}$. Using [20], we write the expansion using Laguerre orthogonal polynomials:(19)$$\begin{array}{cc} F_{R}\left(r\right) & =\frac{r^{2}}{2\sigma_{F}^{2}}e^{-\frac{r^{2}}{2\sigma_{F}^{2}}}\;[{}_{1\;}F_{\;1}\;\left(1;2;\frac{r^{2}}{2\sigma_{F}^{2}}\right)\\ \; & +\sum_{m=1}^{+\infty }\;\frac{{\left(-1\right)}^{m}}{m}\frac{h_{1}^{2m}}{m!}{}_{2\;}F_{\;1}\;\left(-m,-m;1;h_{2}^{2}\right)L_{m-1}^{1}\left(\frac{r^{2}}{2\sigma_{F}^{2}}\right)]. \end{array}$$

Using integration by part, we obtain
(20)$$b_{n}=\lim_{r\to +\infty }a_{n}\left(r\right)F_{R}\left(r\right)-\lim_{r\to 0^{+}}a_{n}\left(r\right)F_{R}\left(r\right)-\int_{0}^{+\infty }\frac{d}{dr}\left[a_{n}\left(r\right)\right]F_{R}\left(r\right)\;dr.$$
Limits $\lim_{r\to +\infty }F_{R}\left(r\right)=1$ and $\lim_{r\to 0^{+}}F_{R}\left(r\right)=0$ exist, and thus
(21)$$b_{n}=\frac{1}{\pi }-\int_{0}^{+\infty }\frac{d}{dr}\left[a_{n}\left(r\right)\right]F_{R}\left(r\right)\;dr.$$
The derivative from the previous equation can be expressed as
(22)$$\frac{da_{n}}{dr}=\frac{n\exp\left(-\frac{r^{2}}{4\sigma^{2}}\right){\left(\frac{r}{\sigma }\right)}^{n-1}}{\sqrt{\pi }\sigma^{3}2^{3(2+n)/2}}\left[8\sigma^{2}\frac{{}_{0\;}F_{\;1}\;\left(;\frac{n+1}{2};\frac{r^{4}}{64\sigma^{4}}\right)}{\Gamma \left(\frac{n+1}{2}\right)}-r^{2}\frac{{}_{0\;}F_{\;1}\;\left(;\frac{n+3}{2};\frac{r^{4}}{64\sigma^{4}}\right)}{\Gamma \left(\frac{n+3}{2}\right)}\right].$$
Using x=r/(2σ) and ([21], 07.17.27.0003.01 ), the derivative expression is shortened to
(23)$$\frac{da_{n}}{dr}=\frac{ne^{-x^{2}}}{2\sqrt{2\pi }\sigma }\left[I_{(n-1)/2}\left(x^{2}\right)-I_{(n+1)/2}\left(x^{2}\right)\right].$$

It can be shown that the derivative is nonnegative, $da_{n}/dr\geq 0,r\geq 0$. Next, we proceed to prove that $F_{R}\left(r\right)$ converges uniformly. We start with Weierstrass *M*-test on series (Equation 19) by finding the supremum over *r*, or $\sup_{x}x^{2}\;e^{-x^{2}}\left|L_{m-1}^{1}\left(x^{2}\right)\right|$. Using ([22], 10.18(14)), ([23], (18.14.8)), we notice that $xe^{-x}\left|L_{m-1}^{1}\left(x\right)\right|\leq mxe^{-x/2}\leq 2m/e$. We can now set the *M* values for the test as $M_{m}=2/e\;{}_{2\;}F_{\;1}\;\left(-m,-m;1;h_{2}^{2}\right)h_{1}^{2m}/m!$. Convergence of the $\sum_{m=1}^{+\infty }M_{m}$ series can further be proven with D’Alembert criterion using $\lim_{m\to +\infty }M_{m+1}/M_{m}$. Because of $\lim_{m\to +\infty }{}_{2\;}F_{\;1}\;\left(-m-1,-m-1;1;h_{2}^{2}\right)/{}_{2\;}F_{\;1}\;\left(-m,-m;1;h_{2}^{2}\right)={\left(1+h_{2}\right)}^{2}$, which can be derived using ([21], 07.23.03.0195.01, 05.03.17.0001.01), it folows that $\lim_{m\to +\infty }M_{m+1}/M_{m}=0$. Now, as $\sum_{m=1}^{+\infty }M_{m}$ converges for all $-\infty <h_{1},h_{2}<+\infty$, the $F_{R}\left(r\right)$ series converges uniformly according to the test, which justifies interchange of integration and summation in computing $b_{n}$ after (Equation 21). Therefore, we continue by defining
(24)$$d_{n,m}=\frac{{\left(-1\right)}^{m}h_{1}^{2m}}{(m+1)!}{}_{2\;}F_{\;1}\;\left(-m,-m;1;h_{2}^{2}\right)\int_{0}^{+\infty }\frac{da_{n}}{dr}\frac{r^{2}\;e^{-\frac{r^{2}}{2\sigma_{F}^{2}}}}{2\sigma_{F}^{2}}L_{m-1}^{1}\left(\frac{r^{2}}{2\sigma_{F}^{2}}\right)\;dr,\;m>0.$$

The integral in the previous equation transforms into
(25)$$\begin{array}{cc} \; & \int_{0}^{+\infty }\frac{da_{n}}{dr}\frac{r^{2}\;e^{-\frac{r^{2}}{2\sigma_{F}^{2}}}}{2\sigma_{F}^{2}}L_{m-1}^{1}\left(\frac{r^{2}}{2\sigma_{F}^{2}}\right)\;dr\\ \; & \;=\frac{n}{2\sqrt{2\pi }\sigma }\int_{0}^{+\infty }e^{-\frac{r^{2}}{4\sigma^{2}}}\left[I_{(n-1)/2}\left(\frac{r^{2}}{4\sigma^{2}}\right)-I_{(n+1)/2}\left(\frac{r^{2}}{4\sigma^{2}}\right)\right]\frac{r^{2}\;e^{-\frac{r^{2}}{2\sigma_{F}^{2}}}}{2\sigma_{F}^{2}}L_{m-1}^{1}\left(\frac{r^{2}}{2\sigma_{F}^{2}}\right)\;dr. \end{array}$$

We introduce $\alpha =2\sigma^{2}/\sigma_{F}^{2}$ and x=r/(2σ);
(26)$$\frac{n\alpha }{\sqrt{2\pi }}\int_{0}^{+\infty }x^{2}e^{-\left(1+\alpha \right)x^{2}}\left[I_{(n-1)/2}\left(x^{2}\right)-I_{(n+1)/2}\left(x^{2}\right)\right]L_{m-1}^{1}\left(\alpha x^{2}\right)\;dx.$$

In principle, these are solvable in a closed form by using identity
(27)$$\begin{array}{cc} \; & \xi_{k,n}\left(\beta \right)=\int_{0}^{+\infty }x^{2+k}e^{-\beta x^{2}}I_{n/2}\left(x^{2}\right)\;dx\\ \; & \;=\frac{1}{2\;\cdot \;2^{n/2}}\frac{\Gamma (\frac{3+k+n}{2})}{\beta^{(3+k+n)/2}}\;{}_{2\;}{\tilde{F}}_{\;1}\;\left(1+\frac{k+n-1}{4},1+\frac{k+n+1}{4};1+\frac{n}{2};\frac{1}{\beta^{2}}\right). \end{array}$$
In the previous equation, symbol ${}_{2\;}{\tilde{F}}_{\;1}\;\left(a,b;c;d\right)$ stands for regularized hypergeometric function ${}_{2\;}{\tilde{F}}_{\;1}\;\left(a,b;c;d\right)={}_{2\;}F_{\;1}\;\left(a,b;c;d\right)/\Gamma \left(c\right)$. Now,
(28)$$\begin{array}{cc} \; & \frac{n\alpha }{\sqrt{2\pi }}\int_{0}^{+\infty }x^{2+k}e^{-\left(1+\alpha \right)x^{2}}\left[I_{(n-1)/2}\left(x^{2}\right)-I_{(n+1)/2}\left(x^{2}\right)\right]\;dx\\ \; & \;=\frac{n\alpha }{\sqrt{2\pi }}\left[\xi_{k,n-1}\left(1+\alpha \right)-\xi_{k,n+1}\left(1+\alpha \right)\right]. \end{array}$$
The Laguerre polynomials are expressed as $L_{m}^{1}\left(x\right)=\sum_{i=0}^{m}\ell_{i}x^{i}$, and, accordingly, we define
(29)$$\Xi_{m,j}\left(z\right)=\sum_{i=0}^{m}\ell_{i}{\left(z-1\right)}^{i}\xi_{2i,j}\left(z\right).$$
Therefore, we can write
(30)$$d_{n,m}=\frac{n\alpha }{\sqrt{2\pi }}\frac{{\left(-1\right)}^{m}}{m}\frac{h_{1}^{2m}}{m!}{}_{2\;}F_{\;1}\;\left(-m,-m;1;h_{2}^{2}\right)\left[\Xi_{m-1,n-1}\left(1+\alpha \right)-\Xi_{m-1,n+1}\left(1+\alpha \right)\right],$$
except for m=0. This case needs to be examined separately, since the hypergeometric function involved is not of polynomial form, ${}_{1\;}F_{\;1}\;\left(1;2;x\right)=\frac{e^{x}-1}{x}$. Further analysis yields
(31)$$\int_{0}^{+\infty }\frac{da_{n}}{dr}\;\cdot \;\frac{r^{2}\;e^{-\frac{r^{2}}{2\sigma_{F}^{2}}}}{2\sigma_{F}^{2}}{}_{1\;}F_{\;1}\;\left(1;2;\frac{r^{2}}{2\sigma_{F}^{2}}\right)\;dr=\frac{1}{\pi }-\int_{0}^{+\infty }\frac{da_{n}}{dr}e^{-\frac{r^{2}}{2\sigma_{F}^{2}}}\;dr,$$
taking into account that $\lim_{r\to 0^{+}}a_{n}=0$, and $\lim_{r\to +\infty }a_{n}=1/\pi$. Moreover,
(32)$$\int_{0}^{+\infty }\frac{da_{n}}{dr}e^{-\frac{r^{2}}{2\sigma_{F}^{2}}}\;dr=\frac{n}{2\sqrt{2\pi }\sigma }\int_{0}^{+\infty }e^{-\frac{r^{2}}{4\sigma^{2}}}\left[I_{(n-1)/2}\left(\frac{r^{2}}{4\sigma^{2}}\right)-I_{(n+1)/2}\left(\frac{r^{2}}{4\sigma^{2}}\right)\right]e^{-\frac{r^{2}}{2\sigma_{F}^{2}}}\;dr.$$

The solution of the previous integral is already found as $\xi_{-2,n}\left(1+2\sigma^{2}/\sigma_{F}^{2}\right)$, Equation (Equation 27). This enables the following identity:(33)$$\begin{array}{cc} \; & d_{n,0}=\frac{1}{\pi }-\frac{n}{\sqrt{2\pi }}\left[\xi_{-2,n-1}\left(1+\alpha \right)+\xi_{-2,n+1}\left(1+\alpha \right)\right], \end{array}$$
which extends the validity of $d_{n,m}$ to m=0 and covers the required range of values.

Therefore,
(34)$$b_{n}=\frac{1}{\pi }-\sum_{m=0}^{+\infty }d_{n,m}.$$

After finishing the derivation of coefficients in Fourier series representation of the composite received signal phase, we are ready to define the probability of the transmitted symbol being wrongly detected. Following the method based on PDF of the received signal phase [3], we can express the conditional SER as
(35)$$P_{e}\left(\varphi \right)=1-\int_{-\pi /M+\varphi }^{\pi /M+\varphi }p_{\Delta }\left(\delta \right)\;d\delta .$$
The average SER is
(36)$$P_{e}=\int_{-\pi }^{\pi }P_{e}\left(\varphi \right)p_{\varphi }\left(\varphi \right)\;d\varphi .$$
Invoking Tonelli’s theorem once again, the formula can be evaluated as
(37)$$P_{e}=1-\frac{1}{M}-2\pi \sum_{n=1}^{+\infty }\frac{c_{n}b_{n}}{n}\sin\left(\frac{n\pi }{M}\right).$$

Computing the SER using (Equation 37) requires a summation of double series, as $b_{n}$ are themselves infinite series over *m*. Clearly, this requires truncation after a certain number of terms in both directions. In Figure 2, we numerically analyze the precision of such truncation for typical system parameters. It can be seen that the numerical error decays quickly after the number of terms increases over a certain threshold. In this particular example, truncation to 1≤n≤22, and 0≤m≤48 provides precision of about two significant digits.

In the case when Gray coding is used for mapping bits into symbols, the approximate error probability per bit can be calculated based on symbol error probability by using $P_{e\mathrm{ba}}=P_{e}/\log_{2}M$. In the following text, these bit error probabilities for BPSK, QPSK and 8PSK are denoted by $P_{e\mathrm{ba}}^{\mathrm{BPSK}}$, $P_{e\mathrm{ba}}^{\mathrm{QPSK}}$ and $P_{e\mathrm{ba}}^{8\mathrm{PSK}}$, respectively. Here, the novel formula is derived for calculating coefficients $b_{n}$. Although this formula is given in the form of an infinite sum, it enables us to evaluate SER in (Equation 37) in detecting MPSK signals over the TWDP channel taking imperfect reference signal phase recovery into account.

#### 3.1.2. Numerical Approach

After analyzing the signal detection in the presence of imperfect reference phase recovery, the formulas for BER in detecting BPSK and QPSK signals can be, respectively, written in the form of [24]
(38)$$P_{e}^{\mathrm{BPSK}}=\frac{1}{2}\int_{r=0}^{+\infty }\int_{\varphi =-\pi }^{\pi }\mathrm{erfc}\left(\frac{r\cos\varphi }{\sqrt{2}\sigma }\right)p_{R}\left(r\right)p_{\varphi }\left(\varphi \right)\;d\varphi dr,$$
(39)$$P_{e}^{\mathrm{QPSK}}=\frac{1}{4}\int_{r=0}^{+\infty }\int_{\varphi =-\pi }^{\pi }\left\{\mathrm{erfc}\left(\frac{r\cos(\pi /4-\varphi )}{\sqrt{2}\sigma }\right)+\mathrm{erfc}\left(\frac{r\cos(\pi /4+\varphi )}{\sqrt{2}\sigma }\right)\right\}p_{R}\left(r\right)p_{\varphi }\left(\varphi \right)\;d\varphi dr,$$
where erfc(·) is notation for the complementary error function defined as ([18], Equation (7.1.2)).

In order to calculate a numerical value of BER, it is necessary to evaluate twofold integrals in (Equation 38) and (Equation 39). The problem of accurately calculating BER is not to be underestimated, as the procedure includes a double integral over a function that includes an infinite alternating series. Obviously, the first concern is the sum, and it is easily proven that the series converges since terms tend to zero and are decreasing. For computing the PDF involved, we use partial sum and the alternating series remainder theorem [25] to determine the number of summands necessary. The precision of summation is set to the level of $10^{-8}$. As for double integrals, we employ integration using the Cartesian rule in which the cubature abscissas are the Cartesian product of two independent quadrature rules for variables *r* and φ. Since the dimensionality is only of second order, the number of points at which the functions must be computed is not forbiddingly large. We set the goal for computing the values of the integrals to at least two significant digits. Since the number of Gaussian points in quadrature rules is a significant parameter, in Figure 3, we explore this influence on the time required for computing a single BER value for the following parameters: BPSK modulation, K=10dB, Γ=0.8, $\sigma_{\varphi }=30{}^{\circ }$, and SNR = 24 dB, using the Mathematica 13 software package. From Figure 3b,c, we conclude that the least amount of time is taken when we choose the number of points to be 7 in the direction of *r* and 15 in the direction of φ. Applying the numerical integration in this sense requires less than 10 s of processor time for computing BER for the given SNR on a 3 GHz i7 processor. When computing multiple values to show the BER curve for a range of SNR values, parallelization can be used to significantly reduce the required time.

#### 3.1.3. Monte Carlo Simulations

Independently from the approaches based on analytical derivation and numerical integration, the BER values are also estimated using Monte Carlo simulations. Monte Carlo simulations are performed according to (Equation 12) and (Equation 13). The samples of the TWDP fading are generated based on (Equation 3) with parameters given by (Equation 6) and (Equation 7). Samples of a uniformly-distributed random variable are obtained according to the algorithm presented in ([26], p. 340), while a Gaussian random variable is generated using the Box–Muller method ([27], p. 383). The random variables with Tikhonov distribution can be generated by applying the Modified acceptance/rejection method ([27], p. 382).

The criteria for estimating a single BER value are as follows: the simulation terminates if at least $10^{4}$ erroneous symbols are detected, or when a maximum of $2\times 10^{9}$ symbols have been transmitted in total.

### 3.2. Mutual Information

The discrete channel (Figure 1) is described by input and output alphabets, as well as the set of conditional (transition or crossover) probabilities. The input and output alphabets are denoted by $\left\{X\right\}=\left\{x_{1},x_{2},\cdots x_{M}\right\}$ and $\left\{Y\right\}=\left\{y_{1},y_{2},\cdots y_{M}\right\}$, respectively. The probability of an input symbol $x_{i},i=1,2,\cdots M$, is denoted by $P(x_{i}),i=1,2,\cdots M$. The conditional probability, $P\left(y_{j}\;|\;x_{i}\right)$, denotes the probability that symbol $y_{j}$ will appear at the channel output if symbol $x_{i}$ appears at the channel input. Since the conditional probability depends only on the current transmitted symbol, the channel is memoryless.

Mutual (transmitted) information ([15], (2.43)) is defined as a difference between a priori ([15], (2.1)) and a posteriori ([16], p. 98) entropies of the input list of symbols. After some mathematical manipulations, the mutual information can be presented in the form of
(40)$$I\left(X;Y\right)=\sum_{i=1}^{M}\sum_{j=1}^{M}P\left(x_{i}\right)P(y_{j}\;\left|\;x_{i}\right)\log_{2}\frac{P(y_{j}\;\left|\;x_{i}\right)}{\sum_{k=1}^{M}P\left(x_{k}\right)P(y_{j}\;\left|\;x_{k}\right)}\;\left[\frac{\mathrm{bit}}{\mathrm{symb}}\right].$$
It is evident that the mutual information depends on the probabilities of input symbols and crossover probabilities. In our analysis, we assume that probabilities of input symbols are equal. Crossover probabilities depend on channel conditions and process of detection. These crossover probabilities are estimated by Monte Carlo simulations that are described in Section 3.1.2. For estimating a value of crossover probability, $10^{5}$ symbols are transmitted over a channel.

## 4. Results and Discussion

On the basis of definitions introduced in Section 2 and expressions presented in Section 3, we present some numerical and simulation results in this Section and offer some discussions of the results. Figure 4a,b and Figure 5a,b present numerical results evaluated on the basis of Formula (Equation 37) together with Monte Carlo simulation results that verify the correctness of the mathematical derivations. Figure 6a,b present results evaluated by applying numerical integration based on (Equation 38). Although Monte Carlo simulation results are not presented in these figures due to clarity, we claim that they overlap with those obtained by numerical integration. The numerical results related to the mutual information presented in Figure 7a,b and Figure 8 are evaluated by means of Monte Carlo simulations based on explanations from Section 3.2.

A typical influence of average signal-to-noise ratio per bit $\rho_{b}$ on system BER is illustrated in Figure 4a. Increasing $\rho_{b}$ has an effect of decreasing the BER, which is an improvement in system performance. However, the BER decrease is not uniform in the entire range of $\rho_{b}$ shown, but is pronounced for low and moderate values of $\rho_{b}$. In the range of higher $\rho_{b}$ values, we observe that the BER values do not decrease significantly following the increase in $\rho_{b}$. On the contrary, the BER tends towards a constant value, which is called the error floor. The appearance of this floor is directly caused by the phase noise resulting from non-ideal recovery of the reference carrier phase. A particular value of the BER floor is dependant on standard deviation of the phase noise. Increase in the phase noise level directly corresponds to higher BER floors and poorer system performance. The BER floor value cannot be reduced by increasing the signal power, and it can only be influenced by the correct design of the subsystem responsible for estimating the incoming sugnal reference phase.

Figure 4b shows the influence of phase noise level on the BER values during the demodulation of BPSK, QPSK and 8PSK signals. Firstly, it is obvious that BER increases with increasing standard deviation of phase noise. Secondly, the 8PSK modulation format starts to be sensitive to phase noise starting from four degrees of standard deviation. Regarding the QPSK format, the BER value remains unaffected until the standard deviation reaches about 8 degrees, while the BPSK format is insensitive to phase noise below 16 degrees. In other words, the BPSK format is more resistant to the influence of the phase noise. This result is also logical because the areas of decision-making in the BPSK format are wider than those in the QPSK and 8PSK formats, so that the phase noise has less possibility to shift the point from one to another area of decision-making.

The influence of the average power ratio of specular-to-diffuse components (i.e., parameter *K*) is shown in Figure 5a. In the range of large values of $\rho_{b}$, error floor appears. In that range, the phase noise has the dominant effect on system performance and the value of error probability does not depend on the value of parameter *K*, but only on the value of standard deviation of phase noise. However, in the range of low and moderate values of $\rho_{b}$, the effect of the propagation environment conditions is stronger in comparison to effects of phase noise. Also, when the value of the standard deviation is lower, i.e., under conditions when PLL better follows changes in stochastic received signal phase, the effect of parameter *K* is stronger.

Similar conclusions to the previous case can be drawn regarding the influence of parameter Γ on the BER value (Figure 5b). Namely, parameter Γ also has a strong influence on the BER values in the middle SNR range, and almost no effect for strong signal levels, i.e., when the system reaches the BER floor. Also, when the phase noise is stronger and has higher standard deviation, i.e., when the non-ideal extraction dominates, performance is less sensitive to parameter Γ changes, in comparison to the case where the reference phase recovery is more precise.

In general, fading parameters *K* and Γ influence performance in a way that is not directly visible from the model presented. For example, if a system has fixed thermal SNR and phase noise levels, then its performance predominantly depends on these fading parameters. We explore this situation in Figure 6a. We analyze the case of BPSK by fixing the BER at $10^{-5}$, and we calculate the combinations of fading parameters that are required to attain the requested BER performance. These constant BER curves show that an increase in the *K* parameter can be compensated by an increase in Γ along the presented curves. Below the curves and to the right lays the area of improved performance that corresponds to the LOS scenario (larger *K*) and/or single dominant specular component (smaller Γ values). Phase noise shifts the curves towards this direction, indicating that more favorable fading conditions are required when significant phase noise is present. Above the curves and to the left is the region that leads to poorer BER performance, and it corresponds to NLOS and/or balanced specular components.

Figure 6b shows the dependence of BER on $\rho_{b}$ for the BPSK modulation format. Firstly, it should be noted that, as in the case of QPSK (Figure 4a), for large values of $\rho_{b}$, an error floor appears. Secondly, when in our numerical and simulation model, $\sigma_{\varphi }=0{}^{\circ }$ is set, which means that the phase estimation is ideal; the same dependence is obtained as in ([9], Figure 6). Therefore, apart from the fact that the results obtained by analytical approach and numerical integration were verified by independent Monte Carlo simulations, this figure shows that our results are reduced to the results from [9], where the assumption about the ideal estimation of the carrier phase was set.

Figure 7 illustrates the influence of the phase noise on mutual information. When transmitting an MPSK signal, the maximum mutual information value that can be achieved is $\log_{2}M$ (bits/symbol). In order to reach this value of mutual information, the channel should be good enough, i.e., the signal power should be sufficiently large in relation to the noise power. Figure 7a shows the mutual information for the QPSK and 8PSK formats. Firstly, it should be noted that the phase noise affects the mutual information in such a way that its maximum value cannot be reached in the presence of the phase noise regardless of the increase in signal power. For example, if $\sigma_{\varphi }=20{}^{\circ }$, the maximum mutual information values for 8PSK and QPSK are 1.8 bits/symb and 1.75 bits/symb, respectively, and not 3 bits/symb and 2 bits/symb, like it is the case when reference signal phase recovery is perfect. The relative decrease in mutual information is larger for 8PSK than for QPSK. This saturation value of mutual information can be increased only by decreasing the phase noise standard deviation, i.e., by proper design of PLL, and not by increasing the signal power.

In Figure 7b, the SNR is fixed at 30 dB, i.e., the SNR value is fixed when the maximum value of mutual information is reached for all modulation formats, which is 3 bits/symb, 2 bits/symb and 1 bit/symb for 8PSK, QPSK and BPSK, respectively. We observe the sensitivity of this mutual information to the change in the standard deviation of the phase noise. Mutual information for 8PSK begins to decrease as early as $\sigma_{\varphi }=5{}^{\circ }$, while mutual information for QPSK begins to decrease at $\sigma_{\varphi }=15{}^{\circ }$, and mutual information for BPSK becomes sensitive to standard deviation of phase noise at $\sigma_{\varphi }=25{}^{\circ }$. The higher the order of modulation, the stronger the mutual information sensitive to phase noise. As was mentioned, the mutual information for BPSK, QPSK and 8PSK when there is no phase noise under given conditions is 1 bit/symb, 2 bits/symb and 3 bit/symb. However, when phase noise appears and if the standard deviation of phase noise is 20 ${}^{\circ }$, the mutual information for BPSK, QPSK and 8PSK is 1 bits/symb, 1.48 bits/symb and 1.86 bits/symb, respectively.

Depending on the fading conditions in a certain environment, increasing SNR causes transmitted information to also increase towards its limiting value for the specific modulation format, but at varying rates. Moreover, phase noise causes the limiting values to decrease, which introduces another level of complexity into analysis. To address this, we analyze the power penalty that the system incurs due to multipath fading when phase noise is present. Taking the transmitted information limit for a fixed phase noise level as a reference, we calculate the amount of additional power that is required in order to recover certain level of transmitted information in fading conditions. This is shown in Figure 8 for the 8PSK modulation example. From the figure, we can conclude that unfavorable fading conditions corresponding to the case when specular components are not dominant over diffuse components (K=5dB) impose significant power penalties when a less than 10% decrease in transmitted information is allowed. On the other hand, LOS (K=15dB) causes only negligible penalty that is under 1 dB for the considered conditions.

## 5. Conclusions

In this paper, we analyzed the transmission of PSK signals over a TWDP channel in the presence of imperfect reference signal phase estimation. Non-perfect estimation of the incoming signal phase directly causes the appearance of the BER floor. The specific value of this BER floor is reducible only by decreasing the phase noise level, i.e., by improving the received signal phase estimation, while the BER floor is insensitive to a further increase in signal power, as well as changing the values of parameters *K* and Γ. The obtained results showed that parameters *K* and Γ strongly influence BER over the range of moderate SNR values. In addition, the results showed that for typical channel conditions, the BPSK modulation format can tolerate phase noise until its standard deviation reaches 16 degrees, while in the same environment, the QPSK/8PSK format is resistant to this phenomenon only up to the standard deviation of eight/four degrees. The results illustrated that the maximum value of mutual information cannot be reached in the presence of the phase noise regardless of the increase in signal power. The saturation value of mutual information can be increased only by decreasing the phase noise standard deviation, i.e., by proper design of PLL, and not by increasing the signal power. In an illustrative scenario, we showed that the maximum value of mutual information for 8PSK, QPSK and BPSK begins to decrease at $\sigma_{\varphi }=5{}^{\circ }$, 15 ${}^{\circ }$ and 25 ${}^{\circ }$, respectively. The higher the order of modulation, the stronger the mutual information shows sensitivity to phase noise.

These values of standard deviation are the starting condition for the correct design of the phase estimator, i.e., the design of the extraction circuit should be undertaken in such a way that the predetermined phase noise levels in terms of its standard deviation are not exceeded. This standard deviation value is determined based on the tolerable BER (mutual information) value and under given channel conditions.

In this paper, we focused on MPSK because it is a simple example that is very close to practical applications. We emphasized that the solution for M=4 directly corresponds to the case of the 4QAM modulation format proposed for modern 5G systems. Our solution for Fourier coefficients in composite signal phase expansion will be useful for all propagation environments that can be described by the TWDP model. In further work, our attention will be focused on finding a general solution for MQAM formats as well. We will improve the model of the receiver part for reference signal recovery, and will consider recovery circuits based on signal squaring and maximum a posteriori estimation.

## Figures and Tables

**Figure 1 entropy-25-01341-f001:**
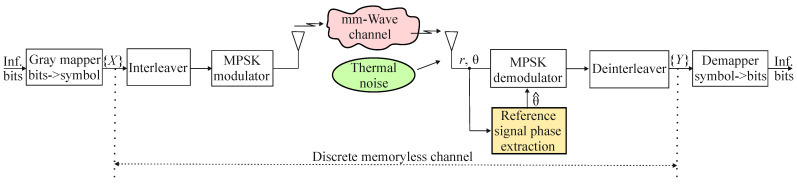
System model.

**Figure 2 entropy-25-01341-f002:**
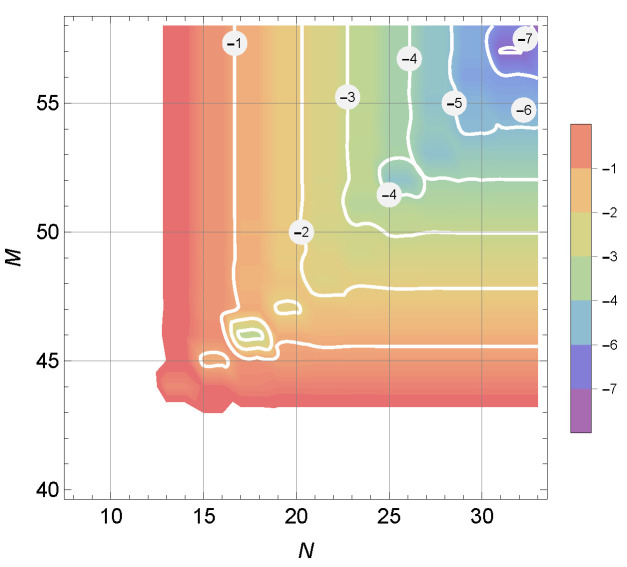
Relative truncation error magnitude during computation of a SER value for typical system paramters: BPSK, K=10dB, Γ=0.5, $\sigma =10{}^{\circ }$, SNR=20dB.

**Figure 3 entropy-25-01341-f003:**
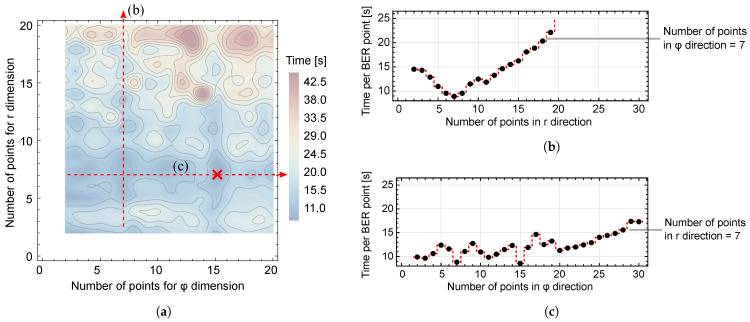
(**a**) Time required for computing BER for a given SNR, showing dependence on the number of Gaussian quadrature points in *r* and φ directions; (**b**) Fixed number of points is seven in the φ direction; (**c**) Fixed number of points is seven in the *r* direction.

**Figure 4 entropy-25-01341-f004:**
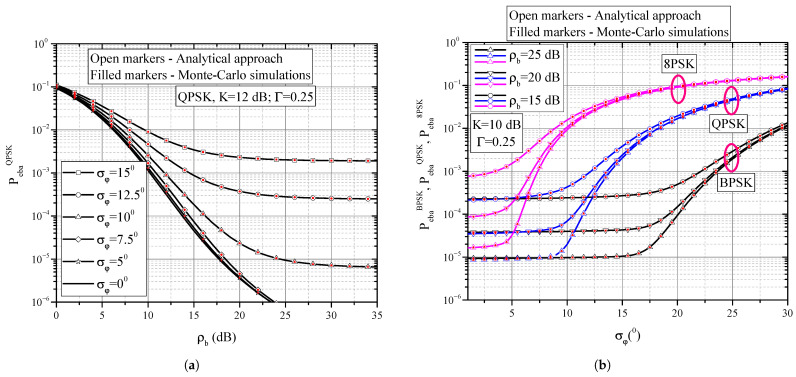
(**a**) BER performance of QPSK receiver for different values of standard deviation of phase noise; (**b**) BER performance of BPSK and QPSK modulation formats in the presence of phase noise.

**Figure 5 entropy-25-01341-f005:**
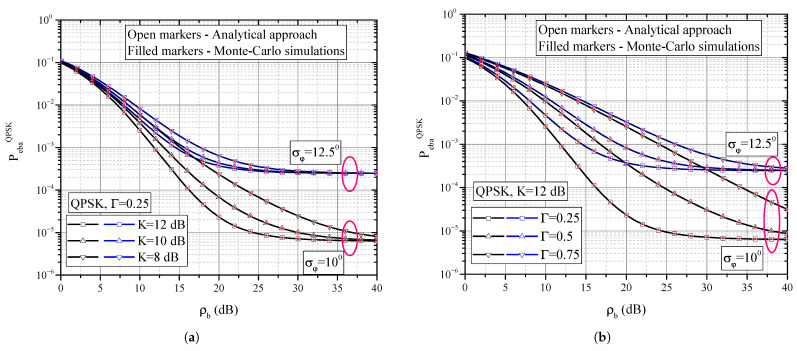
BER performance for different values of fading parameters. (**a**) BER dependence on parameter *K*; (**b**) BER dependence on parameter Γ.

**Figure 6 entropy-25-01341-f006:**
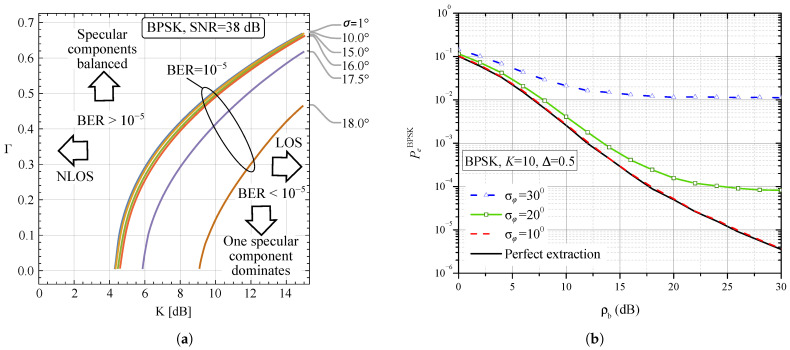
Influence of imperfect reference signal phase recovery on BER performance. (**a**) Different fading conditions: Γ vs. *K*, constant BER curves (BPSK) for fixed SNR, and phase error as parameter. (**b**) Different phase noise levels. Black line corresponds to [9].

**Figure 7 entropy-25-01341-f007:**
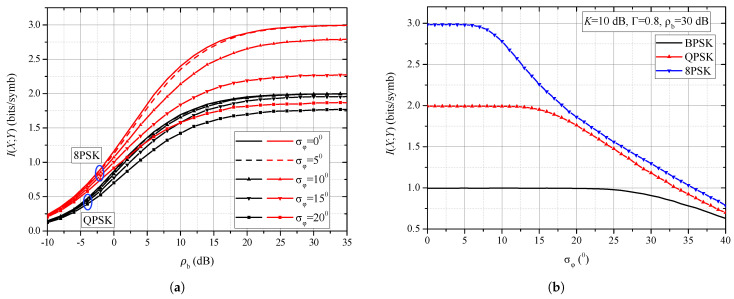
(**a**) Mutual information performance of BPSK and QPSK receiver for different values of phase noise standard deviation. (**b**) Mutual information performance of QPSK and 8PSK modulation formats in the presence of phase noise.

**Figure 8 entropy-25-01341-f008:**
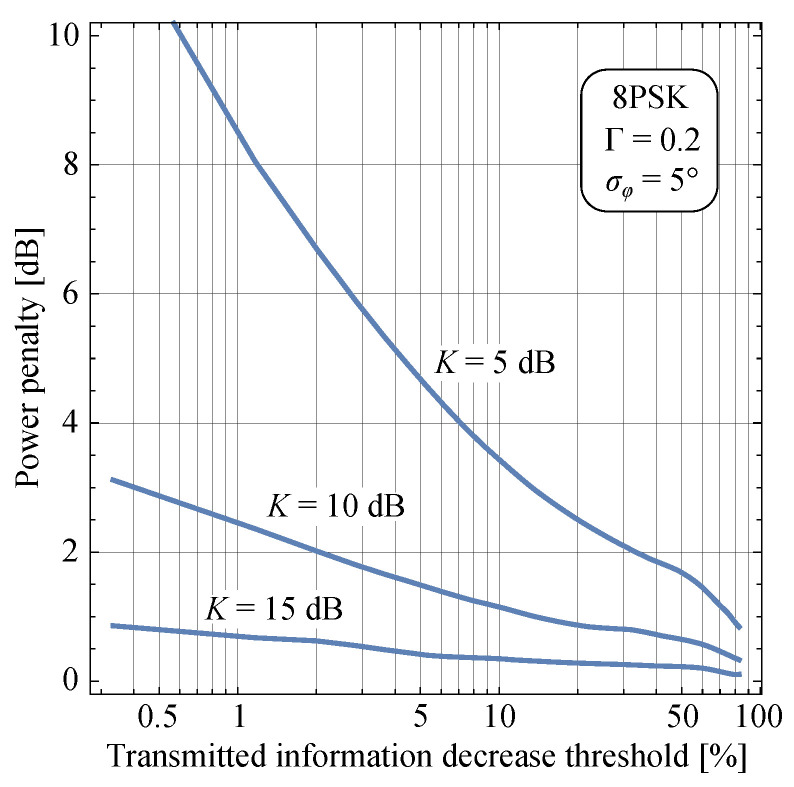
Power penalty versus transmitted information decrease threshold for 8PSK.

## Data Availability

Not applicable.

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
