# Peer review of "On the Effect of Imperfect Reference Signal Phase Recovery on Performance of PSK System Influenced by TWDP Fading"

_entropy, 2023, doi:10.3390/e25091341_

Round 1
Reviewer 1 Report (New Reviewer)
The paper is original, technically sound and clearly presented. It surely merits to be published.
The manuscript addresses an important problem of a phase shift keying signal transmission over a Two-Wave Diffuse-Power channel in the presence of an imperfect reference signal phase estimation, and its impact on the bit-error rate and the quality of transmission parameters. The topic is original and highly relevant in the field, and it addresses a specific gap in the field. The presented results make significant advances as compared with the state-of-the-art and other published material. The methodology is clearly presented and fully relevant to the problem investigated. The conclusion section is fully consistent with the presented results and their analysis in the paper, and it appropriately addresses the main problem defined in the paper. The references are up to date and properly selected. Minor issues: The Authors are encouraged to carefully check the whole manuscript for possible linguistic improvements. For instance, at Page 4: 'Namely, This model of fading' should be 'Namely, this model of fading' etc.Minor issues: The Authors are encouraged to carefully check the whole manuscript for possible linguistic improvements. For instance, at Page 4: 'Namely, This model of fading' should be 'Namely, this model of fading' etc.
Author Response
The authors would like to thank Editor for handling the manuscript, as well as Reviewers for reading our manuscript and submitting objective comments. The comments and suggestions were valuable for us to enhance explanations and improve the quality of the manuscript.
The response of the authors is attached as a pdf file.

Reviewer 2 Report (New Reviewer)
The overall impression from the presented text is quite satisfactory. The text itself is well-written and well-formatted. The authors’ logic and all the derivations are clear and somewhat easy to follow. The obtained results are somewhat interesting.
I see only several issues. Their solution can sufficiently strengthen and improve the paper.
1. Several times throughout the paper, the authors interchange the order of the summation and integration. This is not always allowed and can lead to false results. Thus, such interchanges must be theoretically justified.
2. The expression for the bit error (i.e., (37)) and some others are given in terms of infinite series. Moreover, the derived coefficients (e.g., bn) are the infinite series itself. So there is a two-fold infinite summation. Clearly, they must be truncated with some number of terms inducing a residual error. The authors must study this truncation and the number of terms required for a prescribed accuracy and precision.
3. The authors state that they transmit 104 symbols (for BER evaluation) and repeat it at least 104 times (see line 265). But the plots contain points as low as 10-6, how can this be possible?? Moreover, the curves at these levels (10-5-10-6) are very-very smooth, which is very suspicious.
4. In line 338, it is stated that a threshold BER level of 10-3 was chosen. But for modern communication standards, such an error rate is intolerably high. It would be better to set at least 10-5
5. The authors assume mainly PSK modulation with M up to 8, which is quite unusual for modern telecommunications. Thus, a justification of 8-PSK practical application is expected.
6. Subsection 1.1 is better to name “Literature review”, not just “Literature”.
Author Response
The authors would like to thank Editor for handling the manuscript, as well as Reviewers for reading our manuscript and submitting objective comments. The comments and suggestions were valuable for us to enhance explanations and improve the quality of the manuscript.
The response is given in the attached pdf file.

Round 2
Reviewer 2 Report (New Reviewer)
I'd like to thank the authors for their careful revision of the original submission, which has been sufficiently improved.
This manuscript is a resubmission of an earlier submission. The following is a list of the peer review reports and author responses from that submission.
Round 1
Reviewer 1 Report
The work is focused on a promising topic. The authors present a typical paper where theory from other sources represented by several relationships is combined to develop a description of a more complex system. The simulations are not verified by any experiment or compared with other similar results. The use of simplifying conditions to obtain analytical equations can sometimes lead to difficulties in finding a connection with a real system. In the text, it is necessary to find in several places what type of communication the analyzed system is designed for. This should be clearly stated in the introduction of the paper.
Authors should take an attitude to the following comments and questions:
1. What is the motivation for writing this paper?
2. It is known that the TWDP model is regarded as one of the most promising models for the description of small-scale fading, but it should be stated for which specific communication scenario (LOS, NLOS, free space, urban environment, high mobility, ... ) the channel model described in paragraph 2.2 (including its parameterization) is suitable.
3. It would be very appropriate to determine the difference between simulations of signal phase estimation on the bit-error probability and mutual information for LOS and NLOS scenarios.
4. Does the introduction of relation (5) affect the universality of the model?
5. In the Results and Discussion section, the authors only comment on the simulated results, but do not discuss the physical reasons that lead to them. E. g. on line 265 it is stated that "the influence of the parameter K on the BER value is strong in the range of moderate values of the gamma", but it is not explained or discussed why.
Although the publication contains a lot of work, in my opinion the contribution of the publication is quite small. Overall, I recommend more to connect the theory with a real system.
There are several formal and grammatical errors in the text:
Line 11: affect -> affects.
Line 16: trans-mission -> transmission. Similar mistakes are also on lines 21, 182, 206, 208, 293.
Line 75: focus -> focusing.
Line 77: follows -> follow.
Line 135: missing reference to the values 0 dB to 15 dB.
Relation (8): missing description of Im.
Line 177: presente -> presence.
Relation 13: missing reference to equation.
Lines 217, 218: quantities xi and yi are not formatted with the italics style.
Section 4: average signal-to-noise power uses a different variable in the text and figures.
Line 278: be have -> ?
Line 280: Figure refmutual-information-vs-snr -> Figure 5.
Author Response
Please, see the attachment.

Reviewer 2 Report
1. Title of the manuscript "On the effect of phase noise on bit-error rate and mutual information of mm-Wave MPSK system". Authors are suggested to reframe the Title.
2. Authors have mentioned in the abstract that "direct dependence of bit-error rate and mutual information value on the channel parameters, signal power and standard deviation of the phase noise". This is fundamental concept. What is the novelty in this?
3. The title of the manuscript has MPSK, but the results are mentioned for BPSK, QPSK only. Authors should justify.
4. In Fig. 1, "Discrite memoryless channel" should be "Discrete memoryless channel.
5. The section 3.2 has fundamental formulae. What is the novelty in it?
6. Most of the references example 1, 3,4, 13, 14, 18, 19, 20, 21, 22 are not exactly relevant to the topic of the manuscript.
7. The references that are cited in the text are not in the order.
Author Response
Please, see the attachment.

Round 2
Reviewer 1 Report
The edited version of the paper is, in my opinion, much more readable and understandable. The authors sufficiently accepted the comments of the reviewer. I especially appreciate the justification of the motivation and the much more comprehensive discussion of the results. All formal errors are also corrected (except for the formulation on line 194, where I think it should be "in the presence"). In this form, I think the paper can be published.
Reviewer 2 Report
Section 3.1 has Fundamental definitions.
TWDP model and its analysis is already existing in the Literature.
The authors are strongly recommended to propose a new channel model instead of presenting the exiting models.
The results of the proposed model should be compared with results mentioned in References 5, 9, 11, 12.
Overall, The novelty and authors contributions in the proposed work are very very low.